# Development and Validation of a Model Based on Vegetation Indices for the Prediction of Sugarcane Yield

**Julio Cezar Souza Vasconcelos** [1], **Eduardo Antonio Speranza** [2], **João Francisco Gonçalves Antunes** [2], **Luiz Antonio Falaguasta Barbosa** [2], **Daniel Christofoletti** [3], **Francisco José Severino** [3] and **Geraldo Magela de Almeida Cançado** [2,*]

1    Fundação de Apoio a Pesquisa e ao Desenvolvimento—FAPED, Sete Lagoas 35700-039, Brazil
2    Embrapa Digital Agriculture, Campinas 13083-886, Brazil
3    Cooperativa de Plantadores de Cana do Estado de São Paulo—Coplacana, Piracicaba 13425-000, Brazil
*    Correspondence: geraldo.cancado@embrapa.br

**Abstract:** Currently, Brazil is the leading producer of sugarcane in the world, with self-sufficiency in the use of ethanol as a biofuel, as well as being one of the largest suppliers of sugar to the world. This study aimed to develop a predictive model for sugarcane production based on data extracted from aerial imagery obtained from drones or satellites, allowing the precise tracking of plant development in the field. A model based on a semiparametric approach associated with the inverse Gaussian distribution applied to vegetation indices (VIs), such as the Normalized Difference Vegetation Index (NDVI) and Visible Atmospherically Resistant Index (VARI), was developed with data from drone images obtained from two field experiments with randomized replications and four sugarcane varieties. These experiments were performed under conditions identical to those applied by sugarcane farmers. Further, the model validation was carried out by scaling up the analyses with data extracted from Sentinel-2 images of several commercial sugarcane fields. Very often, in countries such as Brazil, sugarcane crops occupy extensive areas. Consequently, the development of tools capable of being operated remotely automatically benefits the management of this crop in the field by avoiding laborious and time-consuming sampling and by promoting the reduction of operation costs. The results of the model application in both sources of data, i.e., data from field experiments as well as the data from commercial fields, showed a suitable level of overlap between the data of predicted yield using VIs generated from drone and satellite images with the data of verified yield obtained by measuring the production of experiments and commercial fields, indicating that the model is reliable for forecasting productivity months before the harvest time.

**Keywords:** digital agriculture; inverse Gaussian distribution; RPAS; *Saccharum officinarum* L.

## 1. Introduction

Sugarcane crops (*Saccharum officinarum* L.) in Brazil play a major role in agriculture because of their enormous contribution to the production of biofuel and sugar, as well as the growing importance of sugarcane bagasse in new sectors, such as the generation of bioelectricity and the supply of renewable raw material (Vandenberghe et al. [1]). Brazil has been the world's largest producer of sugarcane for decades (Cursi et al. [2]), and a key reason for this success has been the fast and continuous adoption of novel technologies by farmers and industries associated with the sugarcane production in recent decades.

According to the National Supply Agency-CONAB, a Brazilian governmental agency, the sugarcane production in the 2022/23 agricultural season reached a total of 598.3 million tons, which was 3.9% higher than the production in the last season in 2021/22, despite the occurrence of severe droughts and heat in the last years. Thus, as stated above, Brazil has a notable agronomic scenario to develop technologies applied to this crop.

It is well known that sugarcane crops cover large areas in Brazil, making it challenging to evaluate crop performance in real-time by using traditional monitoring tools. Consequently, the use of modern approaches based on aerial images obtained either by satellite or by remotely piloted aircraft systems (RPAS) may be a viable option for surveying vast portions of lands cultivated with sugarcane in an inexpensive and rapid manner. Variations in aerial images of the canopies of cultivated areas are typically tracked by using vegetation indices calculated from data of wavelength bands collected during different stages of crop development, at different locations, and in cultivated fields adopting different agricultural practices. Therefore, slight variations in growth conditions can be identified, compared, and correlated by applying mathematical models. These vegetation indices are typically obtained using mathematical combinations among spectral band data from proximal, suborbital, and orbital sensors that are related to the quantity and development stage of vegetation in the area where the spectral measurement was collected (Hoffman and Todd [3]; Simoes et al. [4]).

For this reason, vegetation indices have been increasingly used in studies to characterize biophysical parameters, such as leaf area and green biomass, which indicate the presence and condition of vegetation at a given time due to their strong correlation with absorbed solar light [5]. Thus, these indices can differentiate vegetated areas from nonvegetated areas, as well as identify stages of the phenological cycle of a specific agricultural crop [6]. Ref. [7] used the NDVI (normalized difference vegetation index) to determine the effect of soil chemical attributes, soil type, and rainfall on cotton yields and yield variability, while [8] evaluated water potential and NDVI variability in four plant species growing in a saline semiarid ecosystem in southern Sonora, Mexico. For the estimation of sugarcane yield, Ref. [9] developed a specific model which uses vegetation indices based on RGB image, digital elevation model, and digital surface model obtained from a camera installed in RPAS. According to the results obtained by these authors, the model had high accuracy for sugarcane fields in Thailand. Furthermore, [10] used vegetation indices obtained from multispectral cameras coupled to RPAS to estimate sugarcane yield at the planting row level during three different growth stages in fields located in Brisbane, Australia. These authors used a generalized linear model, where the results showed that the best predictions occurred during the growth stage.

In this context, classical linear regression models are widely used in various fields of study (Cho et al. [11]; Liu et al. [12]; and Chaboun et al. [13]) and linear regression can accurately identify cause-and-effect relationships among variables of interest. However, some dataset scenarios must be treated more carefully, for example, when explanatory variables have a nonlinear effect on a variable response. In this case, the application of a semiparametric model should be more appropriate.

Indeed, assuming that not all co-variates exhibit a linear effect associated with functional and arbitrary dependence on other explanatory variables, alternative approaches beyond parametric regressions should be taken into consideration. Ref. [14] proposed semiparametric generalized linear models in which the linear predictors were assumed to be of the additive semiparametric type. Further, Ref. [15] proposed a generalized additive model (GAM) that combines the technical features of generalized linear models with additive models while [16] combined a nonparametric regression with parametric regression into a semiparametric regression. In addition, Ref. [17] developed a generalized additive model for location, scale, and shape (GAMLSS) sufficiently flexible to enable the simultaneous introduction of an extensive variable of parameters. As exemplified below, several authors have applied semiparametric models within this context: Ref. [18] showed the advantages of GAMLSS for modeling and interpretation of possible nonlinear climatic impacts during the growth of eucalyptus trees; Ref. [19] proposed additive semiparametric models for symmetric distributions; and Ref. [20] used semiparametric and stochastic frontier models to estimate the efficiency of maize production by smallholders in Zimbabwe.

Against this background, the present study aimed to develop a statistical model based on the analysis of data generated from two field experiments (Field A and Field B) cultivated with four commercial varieties of sugarcane, as described in detail in the Section 2.

Data from multispectral images collected by RPAS at the peak of sugarcane development were compared to agronomic data measured directly from these two sugarcane experimental fields to correlate vegetation indices with sugarcane production. Afterward, the model was validated using yield data obtained from commercial sugarcane fields and vegetation indices calculated from the time series of the Sentinel-2 satellite available on the Google Earth Engine platform. In this sense, the hypothesis tested in this study can be summarized as the application of VIs as predictors for sugarcane yield through a model based on semiparametric statistics.

It follows that the main highlights of this study are (1) RGB and multispectral images of sugarcane fields obtained by drones or satellites are an accurate and fast source of data to evaluate sugarcane growth and biomass accumulation; (2) Adjustment of semiparametric regression models applied to vegetative indices allows the accurate prediction of sugarcane traits such as sugarcane yield; (3) The use of remotely collected aerial imagery can be used as an alternative to combine or even replace laborious, expensive, and time-consuming practices in sugarcane fields; and (4) The model of sugarcane yield prediction demonstrated a reliable ability to overlap predicted data with observed data.

## 2. Material and Methods

### 2.1. Field Experiments

In this study, experiments were carried out in sugarcane fields during the agricultural season of 2020/2021. These experiments were installed in two different locations and four varieties of sugarcane were selected based on the trait of the production cycle (Table 1). The first experiment, identified as Field A, was started in March 2020 and ended with the harvest in April 2021. The second experiment, identified as Field B, was started in June 2020 and ended with the harvest in February 2021. The crop management practices carried out in both experiments were similar to those applied in commercial fields, and standard agronomic operations were used for all treatments in both fields.

**Table 1.** Sugarcane varieties identification and production cycle.

| # | Sugarcane Varieties Used in Field A | Sugarcane Varielites Used in Field B | Varieties Cycle |
|---|---|---|---|
| 1 | CTC1007 | CTC1007 | Normal |
| 2 | RB966928 | RB966928 | Early |
| 3 | CV0618 | CV0618 | Early to normal |
| 4 | CV7870 | CV7870 | Normal |

Figures 1 and 2 illustrate the setup of both experiments in the field and the experimental design as it was allocated in the field. A randomized complete block design, with 28 replicates per variety, was used for each field experiment. Each experimental plot in Field A consisted of four 10 m long lines, spaced by 1.5 m (for a total area of 60 m$^2$ per plot), whereas each experimental plot in Field B consisted of four 9 m long lines spaced by 1.5 m (for a total area of 54 m$^2$ per plot). There were 112 plots per field and therefore a total of 224 experimental plots in the study.

The experiment in Field A was set up in the rural area of Tambaú, São Paulo, Brazil ($-21.708543$, $-47.246643$) using sugarcane stalks with an average length of 1.5 m as seedlings for each variety (Figure 1a). The experiment in Field B was set up in the rural area of Piracicaba, São Paulo, Brazil ($-22.773005$, $-47.580135$) using two-month-old plantlets as seedlings for each variety (Figure 1c). In Field B, a drip irrigation system was used to supplement the water supply during the first three months to enable the plantlets to take root and grow (Figure 1d).

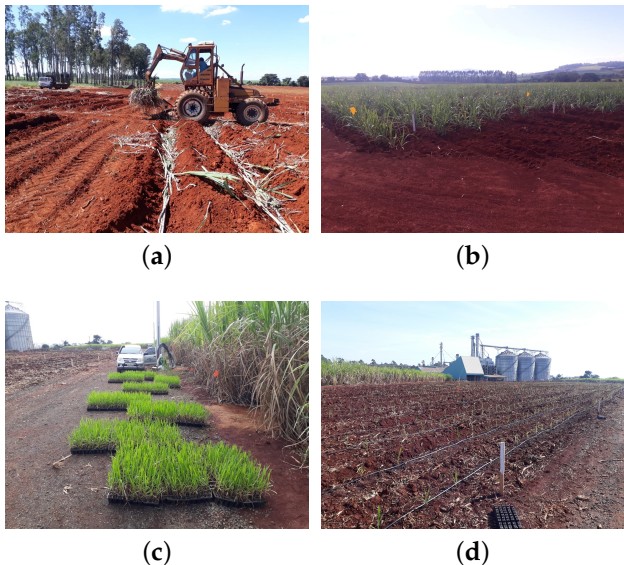

**Figure 1.** The field experiment setup was carried out in two independent areas. (**a**,**b**) Field A. (**c**,**d**) Field B. In Field A cane seedlings were used (**a**) while in Field B plantlets were used (**c**) and drip irrigation (**d**) during the initial stage of development.

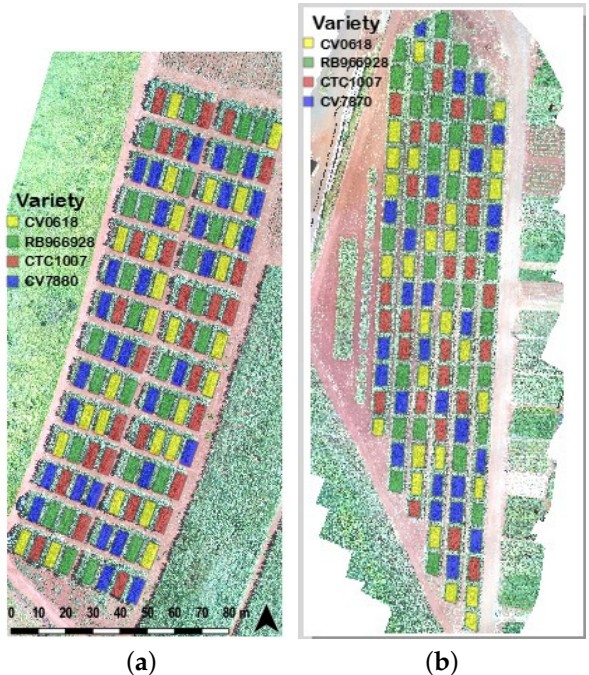

**Figure 2.** Design of experimental Fields A and B and sugarcane varieties randomization. (**a**) Field A, and (**b**) Field B. The orthomosaics were generated by a drone DJI Phantom 4 Pro equipped with RGB and multispectral cameras during the maturation stage.

### 2.2. Vegetation Indices

An ideal vegetation index must reflect small variations in phenological phases during vegetation growth and reduce or even cancel interferences from the type and condition of the soil, the geometry of the incident light at the study site, and atmospheric conditions [21]. The NDVI [22] takes advantage of the difference in the reflectance values in the red wavelength region (between 620 and 640 nm) of the visible spectrum and the near-infrared region (between 720 and 740 nm) and is calculated using Equation (1):

$$NDVI = \frac{\rho_{NIR} - \rho_R}{\rho_{NIR} + \rho_R},$$ (1)

where:

- $\rho_R$ is the reflectance factor of the red band; and
- $\rho_{NIR}$ is the reflectance factor of the near-infrared band.

When the image does not provide multispectral bands, such as those obtained by RGB cameras, the VARI [23] becomes an accessible alternative. VARI is calculated based on reflectance values from the regions of red light (between 620 and 640 nm), green light (between 500 and 565 nm), and blue light (between 450 and 485 nm), using Equation (2):

$$VARI = \frac{\rho_G - \rho_R}{\rho_G + \rho_R - \rho_B},$$ (2)

where:

- $\rho_B$ is the reflectance factor of the blue band;
- $\rho_G$ is the reflectance factor of the green band; and
- $\rho_R$ is the reflectance factor of the red band.

### 2.3. Data Acquisition, Preparation, and Processing

At the end of the sugarcane cycle, the parameter of sugar production identified as Tons of Cane per Hectare (TCH) was determined by weighing each experimental plot during the harvest. The weighing was carried out with the help of a pressure transmitter (VKP-011, Velki, Brazil) coupled to the hydraulic line of the mechanical claws used to collect the bulk of cane stalks harvested per plot. The recorded pressure was then converted from atmospheres (atm) to kilograms (kg) and the weight of each plot was converted to TCH considering the plot area (60 m$^2$ for Field A and 54 m$^2$ for Field B). In the case of the cane evaluation from commercial sugarcane locations used to validate the model, the weighing was performed directly on trucks used to transport the raw material to the mill by using truck scales at the moment of delivery.

The climate data for field experiments A and B were compiled from the NASA Power database as indicated by [24] for the entire period that the sugarcane crop was growing in the field. The dataset was uploaded directly from [25].

For both experimental fields, images were collected in February 2021 (at the peak of biomass accumulation during the phenological stage of sugarcane maturation) using an RPAS Model DJI Phantom 4 Pro, equipped with two cameras: an RGB that could capture images in the visible spectrum; and a multispectral camera adapted to the RPAS, model Sentera Single Sensor NDVI. The second camera had three spectral channels that enabled posterior calculation of the near-infrared and red bands and of the vegetation indices that depend exclusively on these bands [26]. These images were used to generate orthomosaics with a spatial resolution of approximately 2 cm/pixel for calculating the VARI and approximately 5 cm/pixel for calculating NDVI.

The mean values of the NDVI and VARI pixels within each experimental plot were calculated and used in this study (Figure 3) and applied in the model. Further, they were compared to the yield data obtained by the weighing carried out for each experimental plot in the field. The dataset composed of vegetation indices obtained from the orthomosaics generated by RPAS and yield biometrics in the two experimental fields were used only as training and testing stages of the model developement.

The VARI and NDVI indices were also calculated for sugarcane commercial locations used to validate the proposed model, based on satellite images. Section 2.5 describes in detail the procedure used for this calculation.

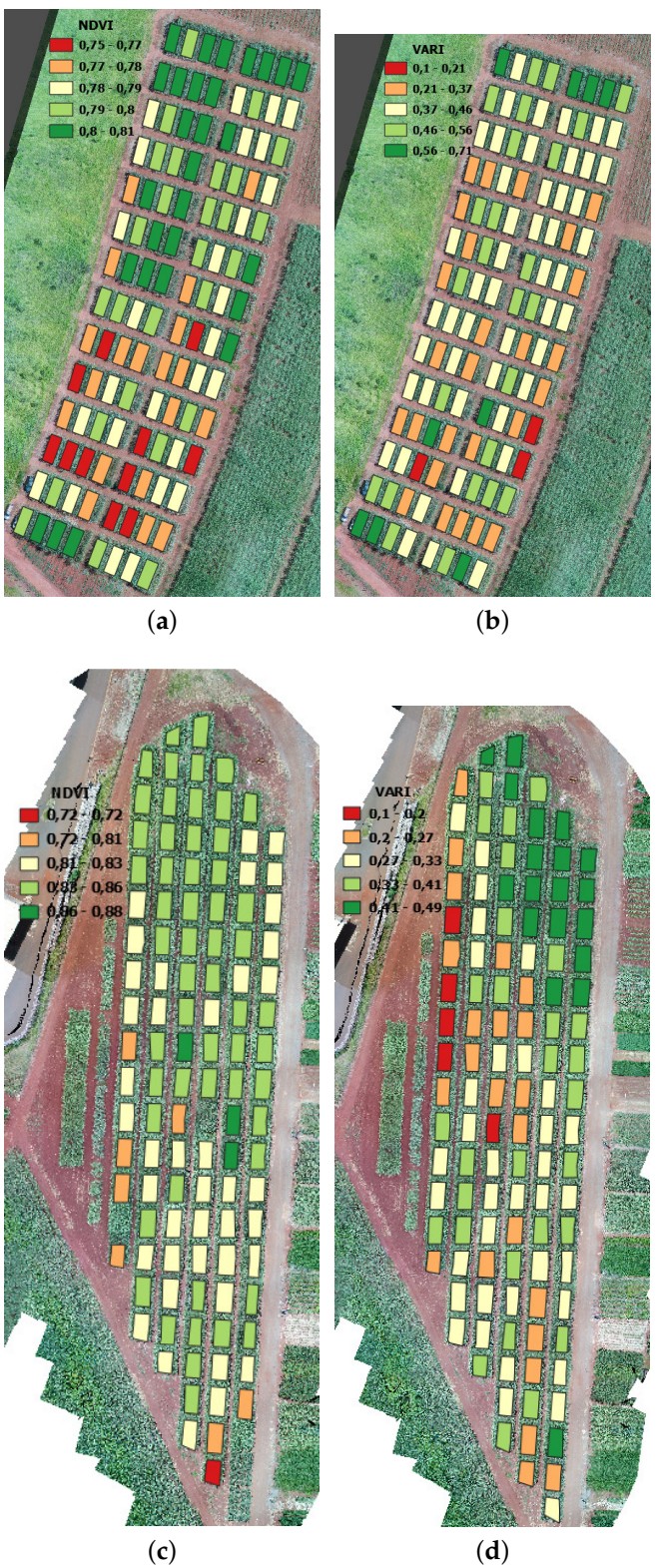

**Figure 3.** RGB orthomosaic images obtained from RPAS in February 2021 and calculated mean of vegetation indices: (**a**) NDVI and (**b**) VARI for Field A; (**c**) NDVI and (**d**) VARI for Field B. The legend shows the scale for polygons representing the mean values of the indices in each plot, where lower values are in red, average values are in yellow and higher values are in green.

*2.4. Statistical Modeling*

The reparameterization technique of [27] was used in this study as follows.

Let $f(y)$ denote a probability density function (pdf) defined as:

$$f(y) = \frac{1}{\sqrt{2\pi\sigma^2 y^3}} \exp\left[-\frac{1}{2\mu^2\sigma^2 y}(y-\mu)^2\right], \tag{3}$$

where $y > 0$, $\mu > 0$ and $\sigma > 0$. Consequently $E(Y) = \mu$ and $Var(Y) = \sigma^2\mu^3$.

The inverse Gaussian semiparametric regression model—We used a semiparametric regression model based on the inverse Gaussian (IG) distribution [27] in this study. We adopted a penalized likelihood function for the IG semiparametric regression model.

Regression analysis is based on criteria for the distribution of $Y_i$ corresponding to a vector of predictor variables, which is denoted by $\mathbf{x}_i = (x_{i1}, \ldots, x_{ip})^T$ and is a vector of covariates that can be categorical or even linear in the response variable. Let $\mathbf{t}_i = (t_{i1}, \ldots, t_{iJ})^T$ be the vector of predictor variables that have a nonlinear effect on the response variable. The systematic component of $\mu_i$ is defined as follows:

$$\mu_i = \exp\left(\mathbf{x}_i^T\boldsymbol{\beta} + \sum_{\xi=1}^{J} f_\xi(t_{i\xi})\right) \qquad i = 1, \ldots, n, \tag{4}$$

where $\boldsymbol{\beta} = (\beta_{11}, \ldots, \beta_{1p})^T$ is the vector of parameters, $f_\xi(\cdot)$ is an unknown arbitrary smooth function related to the predictor variable with a nonlinear effect $(t_i)$ that is controlled in a nonparametric way, for $i = 1, \ldots, n$; and $\xi = 1, \ldots, J$ is the number of predictor variables with nonlinear effects.

Penalized log-likelihood function—Before establishing the penalized log-likelihood function, it is necessary to define the smoothing function. In this study, the $cs(\cdot)$ function is used that is implemented in the `gamlss` [28] package available in open source **R** [29] and based on the cubic smoothing splines function *smooth.spline($\cdot$)*. Note that the $cs(\cdot)$ function does not perform smoothing but uses the attributes in the vector to aid smoothing; for more details see [30].

Thus, the penalized log-likelihood for $\boldsymbol{\theta}$ and $\mathbf{f}$ for the observed sample, is defined as follows:

$$l_p(\boldsymbol{\theta}, \mathbf{f}) = -\frac{n}{2}\log(2\pi\sigma^2) - \frac{3}{2}\sum_{i=1}^{n}\log(y_i) - \frac{1}{2\sigma^2}\sum_{i=1}^{n}\left[\frac{(y_i - \mu_i)^2}{\mu_i^2 y_i}\right]$$
$$- \sum_{\xi=1}^{J}\frac{\lambda_\xi}{2}\mathbf{f}_\xi^T\mathbf{K}_\xi\mathbf{f}_\xi^T. \tag{5}$$

where $\lambda_\xi > 0$ is a smoothing parameter vector that characterizes the smoothness of the curve, that is, it controls the quality of the curve fit. For the smoothed function vector $\mathbf{f}_\xi = (f_\xi(t_{1\xi}), \ldots, f_\xi(t_{q\xi}))^T$, $q$ is the number of distinct and ordered observations of the covariate that is controlled in a nonparametric way, $\mathbf{K}_\xi = \mathbf{Q}_\xi\mathbf{R}_\xi^{-1}\mathbf{Q}_\xi^T$ is a positive definite matrix $q \times q$, where $\mathbf{Q}_\xi$ is an order matrix $q \times (q-2)$, and $\mathbf{R}_\xi$ is an order matrix $(q-2) \times (q-2)$; for more details, see Green and Silverman [31].

Thus, $\theta$ and $f$ are estimated by maximizing Equation (5). We employ the RS estimation algorithm [17] implemented in GAMLSS in the R package [28] for this purpose. The RS algorithm does not use the expected values of the cross derivatives of the log-likelihood but instead maximizes the likelihood function for each parameter ($\mu$ and $\sigma$) in cycles until convergence is reached. Generally, the RS algorithm is stable and fast, and we have not encountered any unforeseen problems regarding convergence and computational cost.

Residual analysis—Residual analysis is a very important tool for verifying whether a considered model is adequate to explain the data. The quantile residuals (qrs) [32] for the IG semiparametric regression model are as follows:

$$\hat{r}_i = \Phi^{-1}(\hat{u}_i),$$

where $\Phi^{-1}(\cdot)$ is the inverse function of the standard normal cumulative distribution and $\hat{u}_i$ is the qrs; more details can be found in gamlss [28].

Comparison of the IG Semiparametric regression model with a multiple regression model—To evaluate the performance of the proposed model developed in this study, the comparison between the results from the IG semiparametric regression model against the results from a multiple regression model, used as standard method, was carried out for the same dataset applied during the model development and model validation. The parameters of $R^2$ and Root Mean Squared Error (RMSE) were calculated for both model approaches and presented in Table 2.

**Table 2.** Estimated quantities for the fitted IG semiparametric regression model adjusted for sugarcane yield.

| Effects | Parameter | Estimate | SE | *p*-Value |
|---|---|---|---|---|
| Intercept | $\beta_0$ | 1.461 | 0.415 | <0.001 |
| Block II | $\beta_{11}$ | −0.003 | 0.025 | 0.899 |
| Block III | $\beta_{21}$ | −0.053 | 0.025 | 0.037 |
| Block IV | $\beta_{31}$ | −0.043 | 0.025 | 0.079 |
| Field B | $\beta_2$ | 0.257 | 0.033 | <0.001 |
| Variety CV0618 | $\beta_{13}$ | −0.077 | 0.025 | 0.002 |
| Variety CV7870 | $\beta_{23}$ | −0.101 | 0.025 | <0.001 |
| Variety RB966928 | $\beta_{33}$ | −0.029 | 0.025 | 0.251 |
| | $\log(\sigma)$ | −4.377 | 0.047 | <0.001 |

### 2.5. Model Validation

Validation is a crucial step for verifying the efficacy, precision, and accuracy of any model. Therefore, the developed model was evaluated using the measured production and the NDVI and VARI indices obtained from satellite images (Sentinel-2 at 10-m spatial resolution) selected during the peak of biomass accumulation for the 30 fields of the six commercial sugarcane locations labeled as A to F, and each location consisted of five fields (see Table S2 and Figures S1–S7 in the Supplementary Materials).

The main difference between commercial fields and experimental fields was the size, since experimental plots used for model development were in a scale of square meters, while commercial locations used for validation were in a scale of hectares.

Similar to the procedure in the experimental stage using drone images, for each field used during the validation, a synthesis image was generated via the Google Earth Engine platform (GEE). The synthesis image of each field is the composition of the maximum value of each pixel contained individually in each of the five plots of every location, considering time-series of Sentinel-2 images, cloud-free, and with radiometric and atmospheric correction, within the period of 30 days during the vegetative peak of the agricultural crop (coincident with the maturation phase).

Given the unavailability of dates provided in the validation dataset, MODIS satellite time series were made available by using the Web application SATVeg (https://www.satveg.cnptia.embrapa.br (accessed on 10 January 2023)). This application was used for inference, in a graphic and visual way, of the time period of the vegetative peak and the sugarcane harvesting for each plot of all locations used for validation. From its respective composite image, for each plot, the mean value of the vegetation index (NDVI and VARI) was calculated, making this new index dataset similar to the dataset used in the modeling with the data from experimental fields A and B (Section 2.3). The TCH data for each of these commercial locations was provided by the farmer/mill as described above.

### 3. Results and Discussion

In many practical applications, the response variable is affected by covariates, making it essential to explore the relationship between the response variable and predictor variables. There are also cases in which one or more covariates may have a nonlinear relationship with the response variable. In these cases, it is necessary to use an adequate model and

incorrect to use the typical linear regression model. Typically, the data fitness is checked by using multiple linear regression models with normal distributions to verify that a normal distribution is suitable for explaining the data. However, this study used an inverse Gaussian (IG) distribution [27] instead of the normal distribution because fitting normal regression models to data generated in this work produced residuals that did not have a normal distribution and thus were in disagreement with the Shapiro–Wilk normality test [33]. Consequently, it did not satisfy the assumption of normality of residuals.

### 3.1. Descriptive Analysis for the Model Development

Figure 4 shows the measured yield of tons of cane per hectare (TCH) for field experiments A and B. Figure 4a shows the average production per sugarcane variety grouped for the two experimental fields, while the Figure 4b shows the average production for each sugarcane variety in each experimental field. Figure 4a also shows that the highest average yields were obtained for the sugarcane varieties CTC1007 and RB966928 and the same result was observed for the productivity of sugarcane varieties in each location separately, as shown in Figure 4b. According to these results, a very similar tendency for the productivity ranking among the sugarcane varieties tested was observed in both experimental fields. These results agree with the concept that the genetic stability of the sugarcane varieties, even when cultivated in different production systems, was not affected, indicating that the experimental design was suitable for the proposed model.

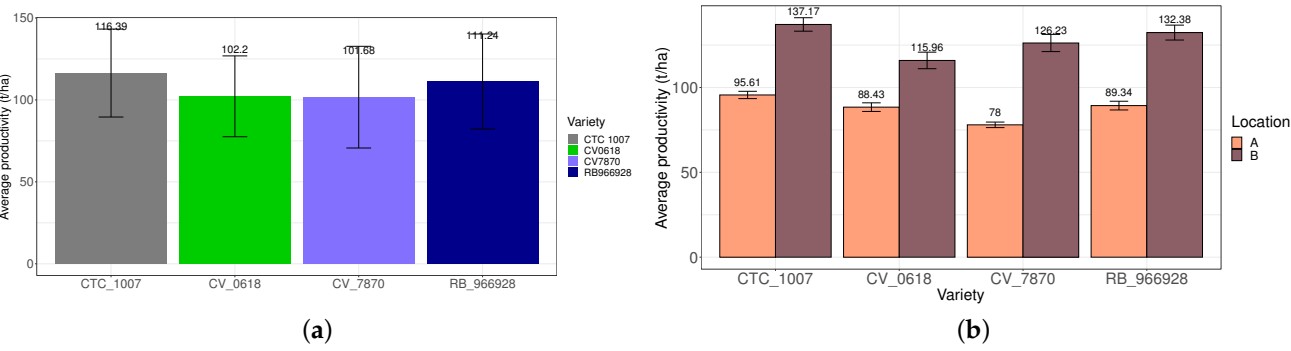

**Figure 4.** Average of sugarcane production in tons of cane per hectare (TCH). (**a**) Graphic for production in Field A and Field B. (**b**) Graphic for the variety production at each experimental field.

The following variables were considered:

- $y_i$: sugarcane yield (t/ha);
- $x_{i1}$: block (I, II, III, and IV). In this case, we need three dummy variables, namely: ($w_{i1}$, $w_{i2}$ and $w_{i3}$);
- $x_{i2}$: local (Field A and Field B); and
- $x_{i3}$: variety (0 = CTC1007, 1 = CV0618, 2 = CV7870 and 3 = RB966928). Here, three dummy variables were also used and defined as follows: ($d_{i1}$, $d_{i2}$ and $d_{i3}$);
- $t_{i4}$: average NDVI; and
- $t_{i5}$: average VARI, for $i = 1, \ldots, 224$.

Here, $x_{i1}$, $x_{i2}$, and $x_{i3}$ are categorical explanatory variables, and $t_{i4}$ and $t_{i5}$ are continuous predictor variables. Figure 5 presents an exploratory analysis of the relationship between the response variable $y_i$ and the continuous explanatory variables $t_{i4}$ (Figure 5a) and $t_{i5}$ (Figure 5b); note that there is a nonlinear relationship between the dependent variable ($y_i$) and the predictor variables. Thus, a semiparametric regression model must be used for a more detailed interpretation of the action of $t_{i4}$ and $t_{i5}$ on ($y_i$).

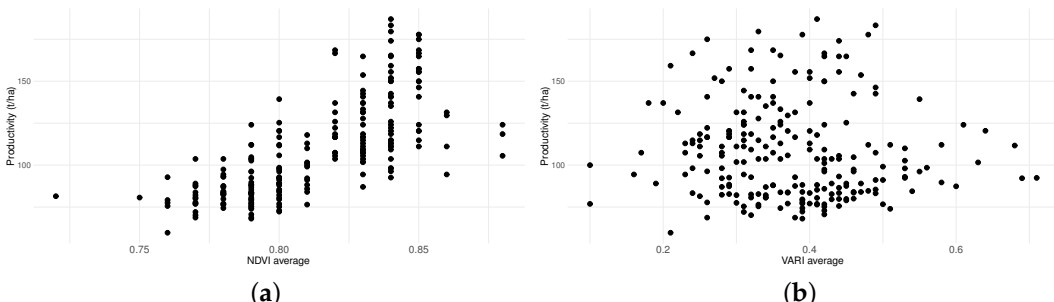

**Figure 5.** Scatter diagrams for sugarcane production, NDVI average versus Productivity data (t/ha), VARI average versus Productivity data (t/ha), from experimental Field A and B during model development. (**a**) Productivity-t/ha ($y_i$) versus NDVI average ($t_{i4}$); (**b**) Productivity-t/ha ($y_i$) versus VARI average ($t_{i5}$).

### 3.2. Results of the IG Semiparametric Regression Model

The inverse Gaussian (IG) is a probability distribution that has been widely used in recent decades to analyze data with right asymmetry. Using degradation data, Wang and Xu [34] proposed a class of inverse Gaussian process models. To solve the multicollinearity problem when present in the inverse Gaussian regression model, Shamany et al. [35] proposed a two-parameter estimator, and demonstrated the excellent performance of the estimator, using a real application of a chemometrics dataset. Recently Kinat et al. [36] proposed the use of control charts based on generalized linear modeling (GLM) when the response variable follows the inverse Gaussian distribution (IG). Meanwhile, Allison et al. [37] presented a new class of goodness-of-fit tests for the inverse Gaussian distribution based on a characterization of the cumulative distribution function.

Therefore, a semiparametric model based on an inverse Gaussian distribution (IG) was developed to predict the sugarcane yield based on VI data. There are three categorical covariates and two predictor variables with a nonlinear effect on $y_i$. Thus, a semiparametric regression model based on the IG distribution was applied with the use of the systematic component (Equation (4)) and the prediction of variables (described in Section 3.1). Figure 6 presents the steps to arrive at the IG semiparametric regression model. From that, the model below was obtained:

$$\mu_i \;=\; \exp\left(\beta_0 + \beta_{11}w_{i1} + \beta_{21}w_{i2} + \beta_{31}w_{i3} + \beta_2 x_{i2} + \beta_{13}d_{i1} + \beta_{23}d_{i2} + \beta_{33}d_{i3} + \sum_{\xi=4}^{5} f_\xi(t_{i\xi})\right).$$

Next, we present the results of the adjusted regression model. Table 3 presents the estimates, with their respective standard errors (SEs) and $p$ values, for the adjusted model. The covariates $x_{i1}$, $x_{i2}$, and $x_{i3}$ are statistically significant at the 5% level, showing that there is a difference between blocks, experimental fields, and varieties in relation to the sugarcane yield. The variables Block I, Field A, and variety CTC 1007 were fixed as intercepts for this analysis.

In terms of the generalized R-squared [38] the model was able to explain 73.8% ($R^2 = 0.7376$) of the data. Figure 7a shows the quantile residuals; it is noteworthy that these residues do not exhibit a trend, proving the adequacy of the model. Figure 7b shows another goodness-of-fit plot, the worm plot of the quantile residuals [39], which validates the model and proves the plausibility of the fitted regression model for explaining the data.

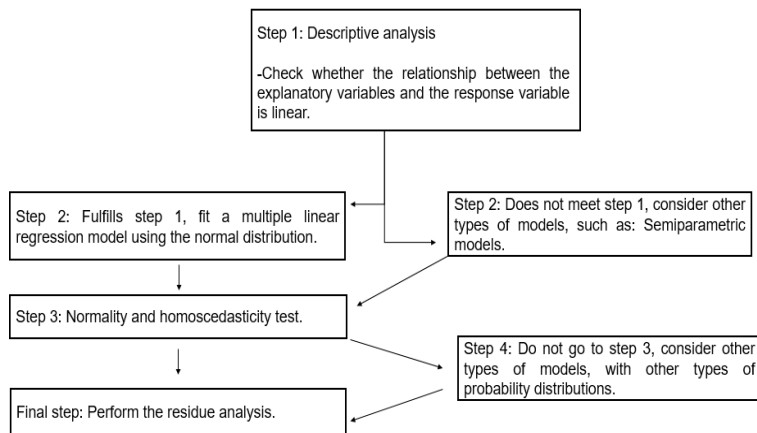

**Figure 6.** Flowchart indicating the main steps carried out during the model development.

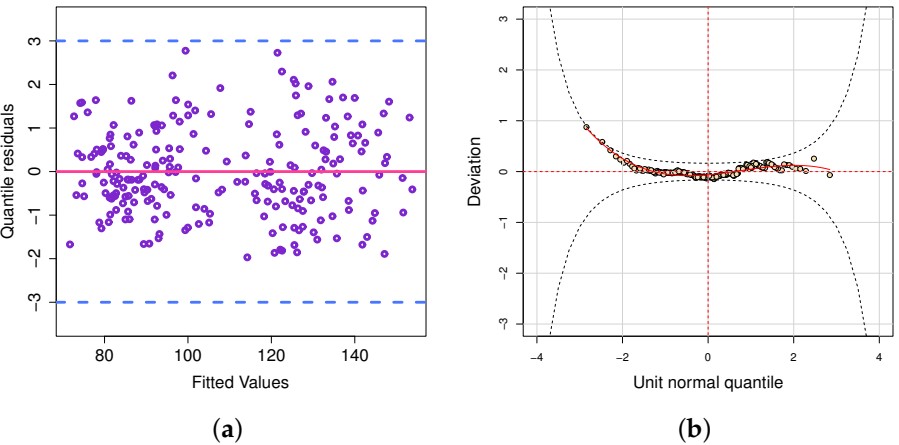

**Figure 7.** (**a**) Plot quantile residuals versus adjusted values; (**b**) Worm plot of quantile residuals.

Table 4 presents the results of the adjusted model for use in comparing the levels of the covariate $x_{i3}$. Note that there are differences among the varieties in terms of the productivity t/ha, with statistically significant differences (at the 5% level) between CV0618 and CV7870 compared to CTC1007 (anchored as intercept). The same result was observed for the production of varieties CV0618 and CV7870 compared with the production of the variety RB966928. These results, obtained by using the IG semiparametric regression model, were determined to be accurate because they agree with the descriptive analysis presented in Table 3. Thus, it can be concluded that the highest sugarcane yields were obtained for the CTC1007 and RB966928 varieties.

**Table 3.** Results of multiple comparisons of the IG semiparametric regression model.

| Hypotheses $H_0$ | Estimate | SE | *p*-Value |
|---|---|---|---|
| CV0618–CTC1007 | −0.077 | 0.025 | 0.002 |
| CV7870–CTC1007 | −0.101 | 0.025 | <0.001 |
| RB966928–CTC1007 | −0.029 | 0.025 | 0.251 |
| CV7870–CV0618 | −0.024 | 0.024 | 0.313 |
| RB966928–CV0618 | 0.048 | 0.024 | 0.045 |
| RB966928–CV7870 | 0.072 | 0.024 | 0.003 |

**Table 4.** Results from the fitted IG semiparametric regression model.

| Effects | Parameter | Estimate | SE | *p*-Value |
|---|---|---|---|---|
| Intercept | $\beta_0$ | 4.653 | 0.685 | <0.001 |
| Location B | $\beta_1$ | −0.036 | 0.050 | 0.491 |
| Location C | $\beta_2$ | −0.562 | 0.042 | <0.001 |
| Location D | $\beta_3$ | −0.386 | 0.048 | <0.001 |
| Location E | $\beta_4$ | −0.284 | 0.047 | <0.001 |
| Location F | $\beta_5$ | −0.535 | 0.153 | 0.005 |
| | $\log(\sigma)$ | −5.034 | 0.129 | <0.001 |

### 3.3. Use of Spectral Indices as Predictors

The NDVI is one of the most well-known and widely used indices because of the ability to reduce soil interference in the vegetation spectral response, atmospheric interference, and the influence of the angle of solar light in multispectral images. The NDVI is not a biophysical measurement but an indirect indication of plant health. However, the NDVI can be influenced by external factors, including weather conditions, agronomic practices, the occurrence of diseases, and nutritional deficits, among several other factors that affect plant performance in the field. The technological evolution of multispectral sensors over the last decade, mainly with regard to radiometric calibrations, has made the use of the NDVI more reliable for agronomic applications [40].

The NDVI vegetation index is calculated using near-infrared to red wavelengths. This index allows the detection of plant canopies in remote sensing data obtained using multispectral cameras. For this reason, the use of NDVI enables easy monitoring of seasonal changes in the development of crops cultivated in the field. The NDVI ranges from −1.0 to +1.0, where negative values typically indicate the presence of water bodies, low values indicate areas with little vegetation, and high values indicate areas with high vegetation vigor. NDVI values for sugarcane can reach saturation peaks of 0.8 approximately during the final stages of growth and maturation because of the NDVI sensitivity to changes in the plant canopy [4].

The VARI, similarly to the NDVI, can be used to enhance the presence of vegetation, mitigate differences in light incidence, and perform corrections of atmospheric effects by using length reflectance values only from the visible spectrum: thus, the VARI can be used to indirectly estimate vegetation fractions with high precision over a wide range of optical wavelengths [41].

In some cases, negative VARI values can be obtained even in the presence of vegetation. The VARI is used to indicate the green level of a specific area in an image and can also be used to detect stress in areas cultivated with sugarcane. The VARI is widely applied to the analysis of RPAS images because the VARI can be easily determined using accessible onboard RGB cameras with sensors that detect only reflectance in the visible spectrum [42].

Although both VARI and NDVI vary between −1.0 and +1.0, the VARI values are typically lower than NDVI values for the same location at the same instant. In general, both the NDVI and VARI are considered indirect indices of detectable variations in the green biomass and leaf area of agricultural crops. In the case of sugarcane, which is based on the production of biomass (leaves and stalks), these vegetation indices could be used as indicators to help estimate the agricultural yield. The development and fine-tuning of mathematical models are required to improve the precision of the estimated yield. The results obtained using these indices could be mathematically interpreted as the relationship function between the response variables associated with the agronomic growth and development of sugarcane varieties influenced by environmental effects [43].

The behavior of predictor variables with nonlinear effects can be observed from the estimated smoothed curves obtained using the IG semiparametric regression model shown in Figure 8. The shaded region represents the 95% confidence bands for the smoothed curve. A practical interpretation of these curves is given below.

1. Figure 8a shows the average NDVI values: the average sugarcane yield increased between NDVIs of approximately 0.70 and 0.85, but then remained constant for an average NDVI close to 0.85.
2. Figure 8b shows the average VARI values: the average sugarcane yield increased between VARIs of approximately 0.25 and 0.60.

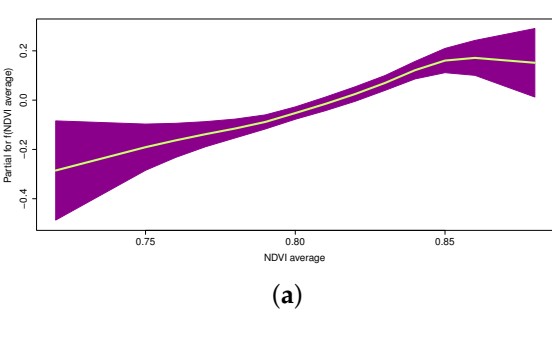

(**a**)

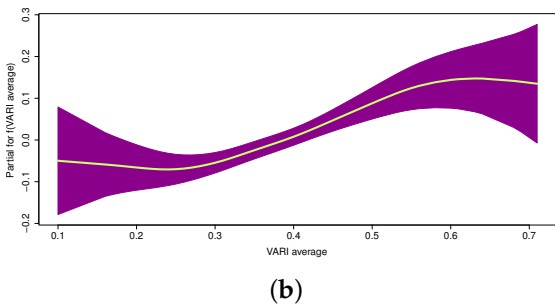

(**b**)

**Figure 8.** The plot of smooth functions fitted to predictor variables. (**a**) Average NDVI ($t_{i4}$); and (**b**) Average VARI ($t_{i5}$). The data for predicted productivity values were obtained from VIs calculated from RPAS images.

Figure 9 shows the predictions of the semiparametric regression model and the observed values. These values refer to the sugarcane yield, and the predicted and observed values are very close, proving that the statistical model is adequate. Thus, this model could be used as an alternative prediction model for future experiments on sugarcane yield.

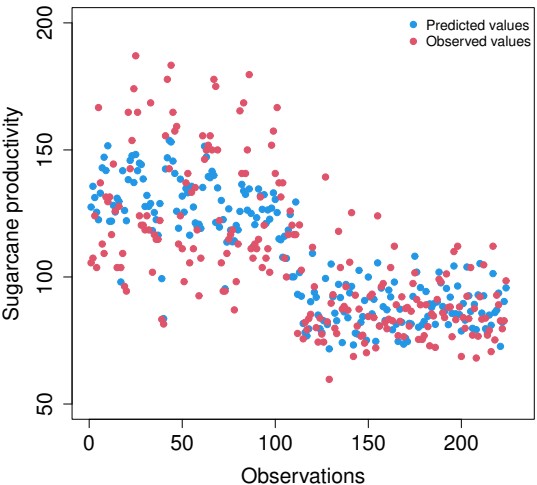

**Figure 9.** Sugarcane productivity plot of predicted values (blue dots) calculated by the model versus observed values (red dots) measured in experimental fields A and B. The data for predicted productivity values were obtained from VIs calculated from RPAS (drone) images.

### 3.4. Climate

Climate, as characterized by short-term weather, is one of the most important variables influencing the success or failure of agricultural activity. Although the climate is a set of complex natural phenomena, two parameters associated with climate are crucial to achieving high productivity. These parameters are soil moisture, which is supplied during the wet season, and temperature, for which the adequate range must be sufficient during crop development. Sugarcane cultivated in Brazil is not typically irrigated. Thus, this crop is highly dependent on rain during the wet season. Heat also plays an important role in the development of sugarcane because sugarcane has a C4-type photosynthetic metabolism that responds to high temperatures. However, both excess and lack of these parameters will impair plant growth and consequently reduce yield.

During the 2020/2021 season, a few months of drought associated with high temperatures occurred in the fields in which the experiments were conducted (Figure 10). The plants developed slowly, very likely because growth was affected by the drought. With subsequent regularization of the environmental conditions, both fields promptly recovered fully, indicating the high resilience of sugarcane. Figure 10 shows that the temperatures reached 40 °C for both Fields A and B during October 2020 and the rainfall rate was less than 20 mm throughout July 2020. These environmental conditions are completely unfavorable for agricultural practice.

Although the climate dataset for the field experiments was obtained from the database of [25] during the whole period that the sugarcane was cultivated, this dataset was not directly involved in the model development. The reason for this decision was that the climate data did not qualify as a quantitative variable for the modeling proposal. The most relevant parameters from the climate data were the averages of maximum/minimum temperatures and the data of monthly accumulated precipitation. The data of these climate parameters from the experimental fields A and B are shown in Figure 10. However, considering each experimental field individually, the climate data were just a constant parameter if compared to the group of explanatory variables in each experimental field. Unlike the climate data that were uniform, the dataset used for the modeling was obtained considering a transverse study. Consequently, in this specific modeling scenario, due to the absence of variability in each experimental field, the climate data did not fit the necessary requirements to be used as an explanatory variable.

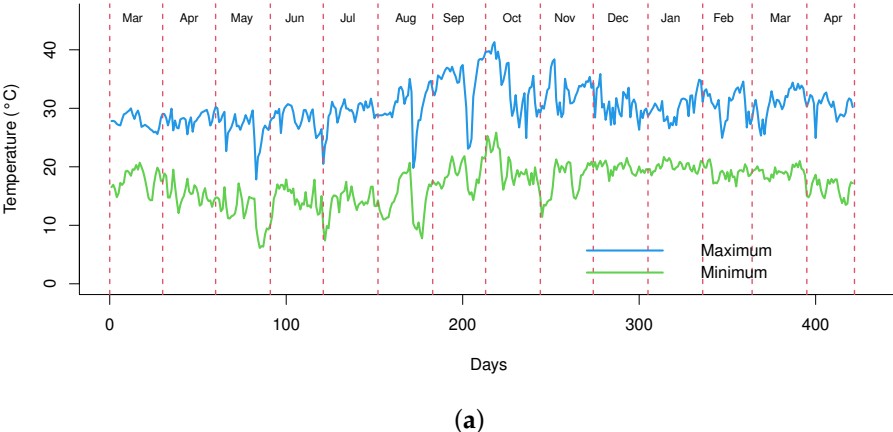

(**a**)

**Figure 10.** *Cont*.

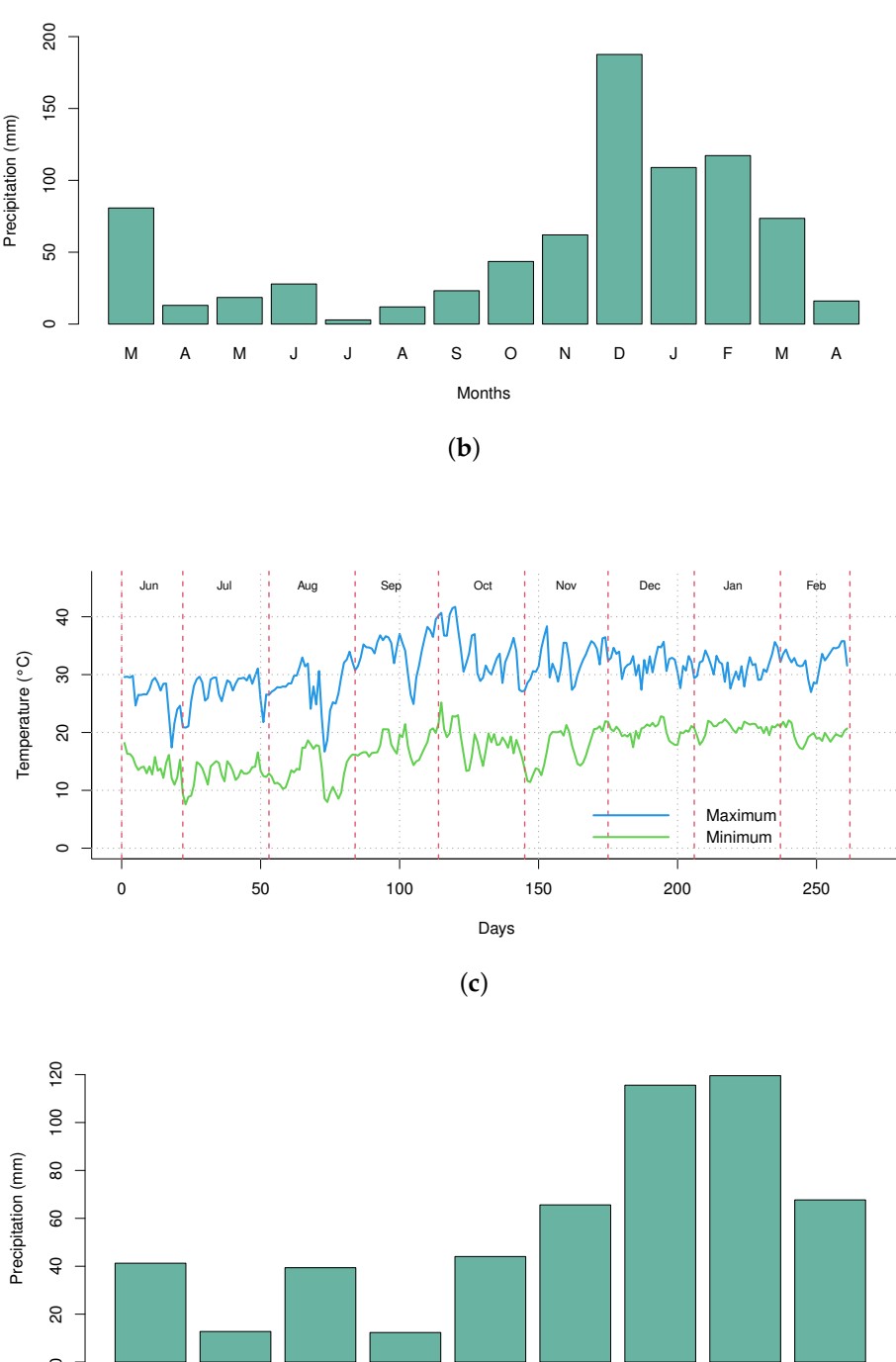

**Figure 10.** Climatology information during the season of 2020/2021. (**a**) Average of maximum and minimum temperature in Field A. (**b**) Monthly accumulated precipitation in Field A. (**c**) Average of maximum and minimum temperature in Field B. (**d**) Monthly accumulated precipitation in Field B. Source (**a–d**) Prepared with data from [25].

## 4. Validating the Model with Data from Commercial Production Fields

### 4.1. Descriptive Analysis of the Validation Data

Note that Figure 11 shows that the highest average productivity occurred in Locations A and B and the lowest average productivity occurred in Location C.

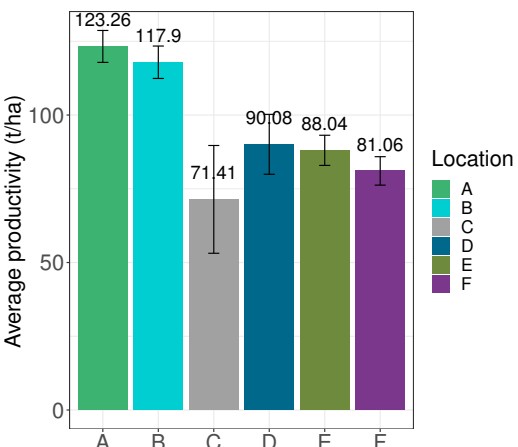

**Figure 11.** Average sugarcane productivity graph (t/ha) for the six locations.

The following variables are defined below:

- $y_i$: average sugarcane productivity (t/ha);
- $x_{i1}$: location (A, B, C, D, E, and F), with five dummy variables, namely ($w_{i1}$, $w_{i2}$, ..., $w_{i5}$);
- $t_{i2}$: area (ha);
- $t_{i3}$: average NDVI; and
- $t_{i4}$: average VARI, for $i = 1, \ldots, 30$.

The Location variable ($x_{i1}$) is a categorical explanatory variable, and the area ($t_{i2}$), average NDVI ($t_{i3}$), and average VARI ($t_{i4}$) are continuous predictor variables. Figure 12 presents an exploratory analysis of the behavior between the productivity response variable ($y_i$) and each continuous explanatory variable. Note there is a clear nonlinear relationship between the productivity response variable ($y_i$) and the continuous explanatory variables, showing that the IG semiparametric regression model needs to be used, as for the data used during the model development.

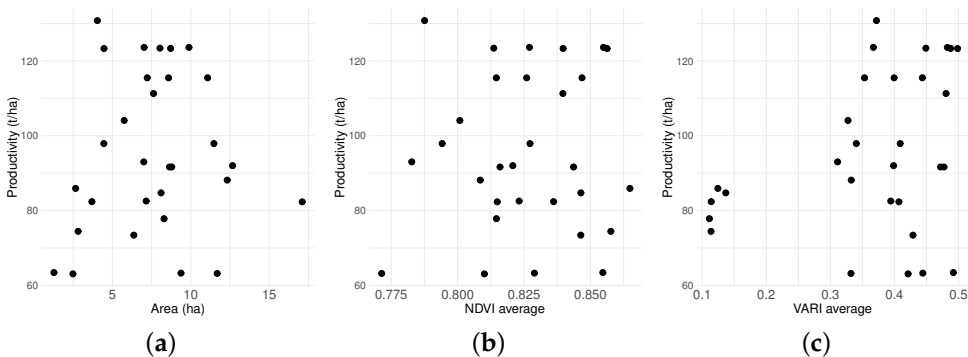

**Figure 12.** Scatter diagrams for sugarcane productivity data (t/ha) from locations A to F used during model validation. (**a**) Productivity-t/ha ($y_i$) versus Area-ha ($t_{i2}$); (**b**) Productivity-t/ha ($y_i$) versus NDVI average ($t_{i3}$); (**c**) Productivity-t/ha ($y_i$) versus VARI average ($t_{i4}$).

### 4.2. Results of the IG Semiparametric Regression Model

Once more, before applying the IG semiparametric regression model to the data, a linear regression model based on a normal distribution was fitted to the data, and the Shapiro–Wilk normality test [33] showed once more that the residual distribution was not normal; therefore, the IG distribution was considered and an IG semiparametric regression model was used. The systematic component of the model is given below.

$$\mu_i = \exp\left(\beta_0 + \beta_1 w_{i1} + \beta_2 w_{i2} + \beta_3 w_{i3} + \beta_4 w_{i4} + \beta_5 w_{i5} + \sum_{\xi=2}^{3} f_\xi(t_{i\xi})\right).$$

Table 5 presents the estimated values with their respective standard errors (SEs) and $p$ values for the fitted model. Note that the covariate $x_i$ is statistically significant at the 5% level, showing that the average sugarcane productivity at Locations C, D, E, and F are different from that of Location Am which was fixed as intercept for this analysis. Later, the reference level of the variable $x_{i1}$ (location) was changed to verify whether the productivities at other locations differ statistically from each other.

For the model validation, the fitted semiparametric model was even more accurate than the data from the model development, since it explained 92.1% ($R^2 = 0.9211$) of the data collected. Figure 13a shows that the quantile residuals are well dispersed and show no trend, proving the fitness of this model. Note that the excellent results for the residuals in Figure 13b, namely the worm plot of the quantile residuals [39], both validated the models and proved the regression model was well-adjusted to the data, as shown by the analysis of the residuals in Figure 7 in Section 3.2.

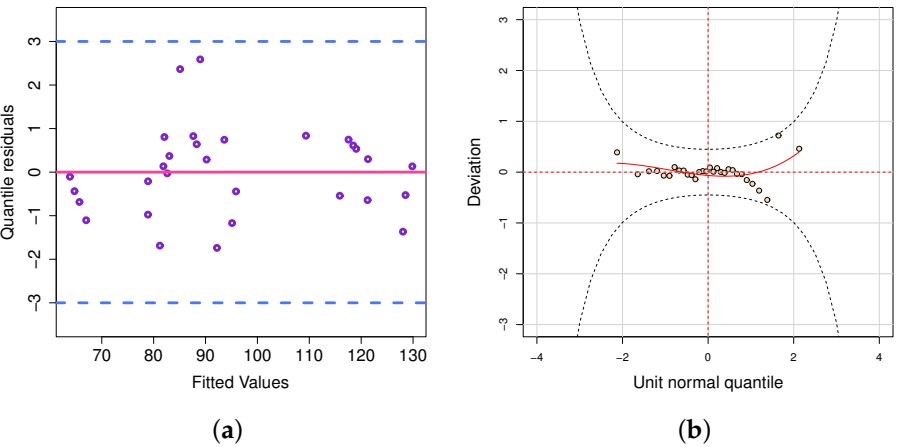

|     |     |
| :-: | :-: |
| (a) | (b) |

**Figure 13.** (**a**) Residual analysis of the IG semiparametric regression fitted; (**b**) Worm plot of quantile residuals.

Table 6 refers to the other levels as a reference in the intercept (in addition to level A which is already in Table 5). To reach results in Table 6, the level B was fixed and compared with the other levels (excluding the level A which was already shown in Table 5). This procedure of taking the variable as an intercept was also applied to the levels C, ..., F. Therefore, Table 6 presents the results of the adjusted model for use in comparing the levels of the explanatory variable location ($x_{i1}$).

It is worth highlighting that levels B,..., F of $x_{i1}$ refer to the dummy variables ($w_{i1}, \ldots, w_{i5}$) described in the Section 4.1. Note that the average productivity of Locations C, D, E, and F differs from Location B, and the average productivity of Locations D and E differs from Location C while the difference observed in the productivity for other Locations are not statistically significant.

**Table 5.** Test of hypotheses for local levels of average sugarcane productivity.

| Hypotheses $H_0$ | Estimate | SE | *p*-Value |
|:---:|:---:|:---:|:---:|
| C–B | −0.526 | 0.047 | <0.001 |
| D–B | −0.350 | 0.046 | <0.001 |
| E–B | −0.248 | 0.044 | <0.001 |
| F–B | −0.499 | 0.139 | 0.004 |
| D–C | 0.176 | 0.039 | 0.001 |
| E–C | 0.278 | 0.045 | <0.001 |
| F–C | 0.027 | 0.144 | 0.853 |
| E–D | 0.102 | 0.046 | 0.051 |
| F–D | −0.148 | 0.132 | 0.285 |
| F–E | −0.250 | 0.149 | 0.123 |

Figure 14 shows the estimated smoothed curves obtained using the IG semiparametric regression model for the predictor variables with nonlinear effect. The following conclusions are drawn from this figure.

1.  Figure 14a shows the area values in hectares (ha). Note that the average sugarcane productivity was highest for areas between approximately 1 and 6 ha but then declined for areas above 6 ha.
2.  Figure 14b shows that the average sugarcane productivity increased between average values of NDVI of approximately 0.75 and 0.80 but remained constant for average NDVI values above 0.80.
3.  Finally, Figure 14c shows that the average sugarcane productivity increased between average VARI values of approximately 0.10 and 0.33, decreased between average VARI values of approximately 0.33 and 0.45, and increased for average VARI values of approximately 0.45 and higher.

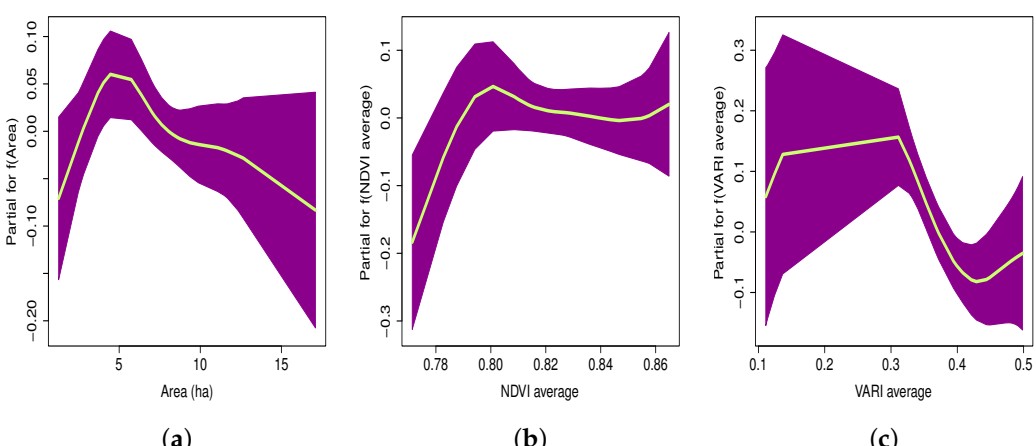

(**a**)                                (**b**)                                (**c**)

**Figure 14.** Plot of smooth functions fitted to predictor variables. (**a**) Area ($t_{i2}$); (**b**) Average NDVI ($t_{i3}$); and (**c**) Average VARI ($t_{i4}$). The data for predicted productivity values was obtained from VIs calculated from satellite (Sentinel-2) images.

Figure 15 shows the predicted values of the IG semiparametric regression model and the observed values. The predicted values are very close to the observed values, proving the plausibility of the statistical model for explaining these data on sugarcane productivity by using vegetation indices applied to different locations.

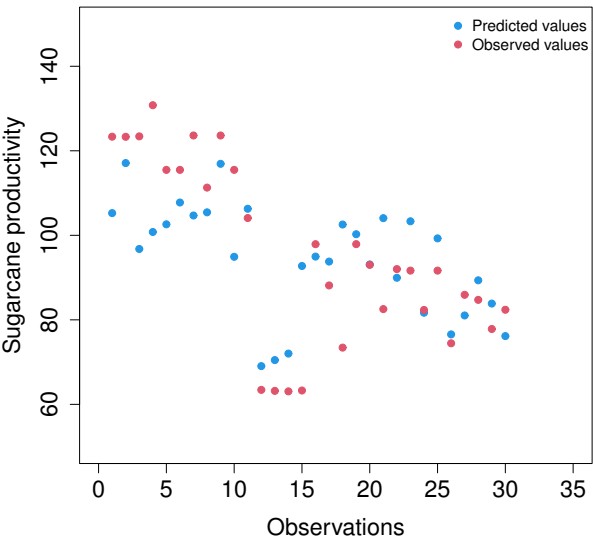

**Figure 15.** Sugarcane productivity plot of predicted values (blue dots) calculated by the model versus observed values (red dots) measured in commercial fields. The data for predicted productivity values was obtained from VIs calculated from satellite (Sentinel-2) images.

### 4.3. Model Comparison

The IG semiparametric regression model was also compared to a multiple linear regression model (standard type of modeling using Inverse Gaussian distribution) which does not consider the smoothing function. Table 2 presents a comparative summary between the two models, and it is possible to notice that the predominance of the IG semiparametric regression model was not only in the values of $R^2$ but also due to the superiority related to the Root Mean Squared Error (RMSE). The RMSE is a measurement that determines the average distance between the model's predicted values and the observed values from a dataset. Actually, the smaller the magnitude of RSME, the better are the fit and predictive ability of the model. In this way, the proposed semiparametric regression model might be suggested as an alternative method rather than a multiple regression model for the sugarcane yield prediction.

**Table 6.** Evaluation metrics of semiparametric regression model IG and multiple regression model for data obtained from field experiments and commercial production.

| Statistical Measures | IG Semiparametric Regression Model | Multiple Regression Model |
|:---:|:---|:---|
| | Field experiment data | |
| $R^2$ | 0.737 | 0.651 |
| RMSE | 16.109 | 16.776 |
| | Commercial production field data | |
| $R^2$ | 0.921 | 0.826 |
| RMSE | 5.998 | 8.657 |

RMSE: Root Mean Squared Error.

Although there is no previous report about the performance using a semiparametric regression model, as with the model developed in this study, against other model approaches using sugarcane yield data, it is worth remarking that [43], using regression random forest models based on satellite, agronomic, and meteorological data, compared five methods for sugarcane yield forecasting. These authors observed that the calibration of these models is specific to a region and the application in localities with different conditions must require the inclusion of more data in the training steps. They also observed that the model based on the combined data of satellite and field could predict the sugarcane yield with the best

accuracy when compared with the other models. Meanwhile, Refs. [44–48] comparing a regression model with a principal component model, observed that the last one generated better $R^2$ values than the regression model for the forecast of sugarcane yield. Therefore, we should take into consideration that the response of a specific model might be intrinsically associated with the condition in which the model is applied and consequently, caution must be taken to prevent spurious comparisons.

Additionally, Figure S6 in the Supplementary Materials shows that the model prediction output was highly related to the measured production for all locations evaluated during the model validation, indicating the assertiveness of this model to predict sugarcane yield.

## 5. Conclusions

The IG semiparametric regression model proved to be suitable for application to experimental data (Section 3) by explaining 73.8% of the observed data based on the generalized R-squared and explaining 92.1% of validation data obtained from commercial fields (Section 4). For the model development, the predictor variables were "fields", "varieties", "NDVI", and "VARI", and the CTC1007 and RB966928 varieties exhibited higher productivity in Field A than in Field B. However, the same effect was not observed for the CV7870 and CV0618 varieties, indicating the effectiveness of this type of approach in correlating data obtained from different sources and in explaining the same phenomenon, in this case, sugarcane productivity. Meanwhile, for the model validation, the predictor variables were "Location", "Area", "NDVI", and "VARI", and the productivity was found to be higher at Locations A and B than at other locations. Finally, in both scenarios (development and validation) described in Sections 3 and 4, the results obtained from the IG semiparametric regression model were adjusted and agreed with the respective descriptive data analysis. The model was also versatile for elucidating the effect of continuous predictive variables ("Locations" or "Experimental Fields", "NDVI", and "VARI") on the response variable of the production (TCH) and was considerably easier to apply than multiple linear regression models that only enable the identification of a positive or negative direction associated with parameter estimation.

A residual analysis proved that the IG semiparametric regression model was well adjusted to the data. That is, the proposed approach was able to satisfactorily model datasets containing continuous predictor variables, including vegetation indices such as the NDVI and VARI.

Additionally, to verify the statistical performance of the IG semiparametric regression model, its versatility was compared to the standard multiple linear regression model (modeling approach usually applied to inverse Gaussian distribution) and which does not consider the smoothing function. The results indicated that the proposed semiparametric model showed a better performance than the standard model.

Assessing field productivity parameters by using conventional methods of direct field sampling can be inaccurate, tiresome, and time-consuming. Thus, this study demonstrated that spectral indices, such as NDVI and VARI that are easily extracted from aerial images collected with drones or satellites, provide a feasible alternative to support or even replace non-automatic parameters. Additionally, results of model validation with on-farm data indicate that the development of such tools can be worthy for daily-basis operations carried out by farmers, improving the agronomic efficiency of their crops.

Thus, this research shows that the use of VIs from images captured by RPAS or satellites can be used to predict sugarcane yield in a reliable way.

**Supplementary Materials:** The following supporting information can be downloaded at: https://www.mdpi.com/article/10.3390/agriengineering5020044/s1.

**Author Contributions:** Supervision, G.M.d.A.C.; The experiments were designed and carried out in the field by G.M.d.A.C., D.C. and F.J.S.; The data were analyzed by J.C.S.V., E.A.S., J.F.G.A. and L.A.F.B. The article was written and reviewed by J.C.S.V., E.A.S., J.F.G.A., L.A.F.B. and G.M.d.A.C. All authors have read and agreed to the published version of the manuscript.

**Funding:** This research was funded by the project Embrapa-Coplacana-Faped, SEG 30.19.90.005.00.00.

**Institutional Review Board Statement:** Not applicable.

**Informed Consent Statement:** Not applicable.

**Data Availability Statement:** To verify the possibility of using the data available in the supplementary material or the images collected in the field, please contact the corresponding author.

**Acknowledgments:** The authors would like to thank Gabriel Bellomi Schiavon from the Schiavon Group for providing the dataset for commercial sugarcane fields and Denize P. dos Santos for assistance with the data analysis.

**Conflicts of Interest:** The authors declare no conflict of interest.

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
