# Peer review of "Development and Validation of a Model Based on Vegetation Indices for the Prediction of Sugarcane Yield"

_agriengineering, doi:10.3390/agriengineering5020044_

Round 1

Reviewer 1 Report

I reviewed the paper entitled "Development and validation of a model based on vegetation indices for the prediction of sugarcane yield" by Vasconcelos et al. Although the topic is interesting, the paper is not well-organized and well-documented. There are not enough new and relevant citations in several parts of the paper, and the lack of suitable references is one concern. The introduction section requires further elaboration to justify the importance and place of this work. There is no explicit section devoted to data explanation, and the provided information is scattered and insufficient. The descriptions of the proposed method are not enough to make it easily reproducible, and some descriptions are given in other subsections that make it hard to understand the procedure fully. The most notable shortcoming of this paper is the results and discussion section. It is hard to follow the provided results, and many questions arise in its different parts, making it difficult to understand. This section also requires a thorough revision and re-organization as an academic paper. Furthermore, the results of the proposed method were not compared to any other previously used method to justify its applicability. Some explanations are missing, and some just come up without prior justification. In light of these, I recommend the rejection of this paper.

Some specific comments are provided below:

1- The introduction section lacks sufficient references. Suitable references are required for sentences in lines 36-39, 57-60, and 74-76.

2- In the third paragraph of the introduction section, the use of vegetation indices in different applications is mentioned. I recommend revising this paragraph and providing more recent papers related to sugarcane studies using vegetation indices, especially relevant to your task. In this case, this paragraph also can highlight the place of your research among previously related publications. You can Provide relevant studies with a few explanations of their workflow.

3- Please make the fifth paragraph of the introduction section more concise. What is the utility of introducing different methods? It is better only to provide the limitations of previous works and how they should be resolved to justify your proposed method.

4- Line 111; The first sentence requires a revision. 

5- What are the codes provided in Table 1?

6- Section 2.1 is "Field experiments"; however, it includes explanations about data and vegetation indices. These explanations should be provided separately. So, add a new subsection for data and vegetation indices.

7- The explanations about data acquisition, data preparation, and processing are not enough. Further explanations are required for both RPAS and Sentinel-2. These explanations are not organized, and it is hard to follow. I recommend introducing the data (both RPAS and Sentinel-2) and providing a thorough explanation of their acquisition and pre-processing, and then providing explanations about the vegetation indices. Also, it is suggested to provide sample RGB images over the sugarcane field using both RPAS (e.g., entire mosaic) and Sentinel-2.

8- Please merge subsections 2.2, 2.3, and 2.4 as "proposed method" subsection.

9- Lines 184-186 and 186-188; Please provide suitable references.

10- Figure 3; Please provide the average value of vegetation indices in each plot. Also, provide a legend so that the colors will be understandable.

11- Line 242; "Figures 15 to 21 and Table 7"; There is no supplementary information to this paper.

12- Section 3; It is better only to conserve results and discussion in this section. Therefore, the description of the analysis should have been clearly stated in the methodology section.

13- It is hard to follow the explanation in Section 3, and many questions arise when reading this section. It requires a thorough revision and re-organization.

14- Figure 5; What are the sources of NDVI and VARI?

15- Line 280; It is not clear how you reached this model. Explanations of each step and each component should have been provided explicitly. 

16- Lines 303-305; Please provide a suitable reference.

17- Lines 356-358; I don't think your data was large (from Figures). Why have you claimed that your model was versatile to handle large data? How many corresponding observations were used?

18- Figure 8; What is the index?

19- Section 3.4; Several explanations are provided about the climatic factors. But these are not enough, and deeper explanations are required about the effect of the climatic variations on your model and results. How do they affect your model?

20- Line 380; What do you mean by "the real know yields"? Did you use any simulated yield before?

21- Line 382; Please provide the location of Six sites. You can show them in the revised Figure 3. What do you mean by "30 fields"? You have only been introduced to fields: Field A and Field B.

22- Section 4.1; Please see comment 12.

23- Line 410; It is not clear how you reached this model. Explanations of each step and each component should have been provided explicitly.

24- Table 4; Where are the results of location A?

25- Line 424; There is no location A in Table 4.

26- The provided explanations and workflow to reach Table 5 are not clear.

27- Figure 14; What is the index?

My suggestion for revising the results section is to divide the results into two subsections. The first should be devoted to the results of the RPAS, and the second should be to Sentinel-2. Further comparisons and discussions should be given after these two subsections. The required information about the equations should have been provided explicitly in the previous subsection.

Author Response

Response to Reviewer 1 Comments

Dear reviewer, first of all, we thank you for your comments and contributions to improving our manuscript. To each reviewed point was added an explanation or justification. All adjustments are indicated in the text with the color red. Therefore, we hope to meet your expectations. 

Comments  - English language and style

Response – The manuscript was edited by AJE (attached follows the certificate of edition) to an adequate English language standard. Besides that we also did a careful reading throughout the text to correct any misspellings or grammar errors.

General overview - I reviewed the paper entitled "Development and validation of a model based on vegetation indices for the prediction of sugarcane yield" by Vasconcelos et al. Although the topic is interesting, the paper is not well-organized and well-documented. There are not enough new and relevant citations in several parts of the paper, and the lack of suitable references is one concern. The introduction section requires further elaboration to justify the importance and place of this work. There is no explicit section devoted to data explanation, and the provided information is scattered and insufficient. The descriptions of the proposed method are not enough to make it easily reproducible, and some descriptions are given in other subsections that make it hard to understand the procedure fully. The most notable shortcoming of this paper is the results and discussion section. It is hard to follow the provided results, and many questions arise in its different parts, making it difficult to understand. This section also requires a thorough revision and re-organization as an academic paper. Furthermore, the results of the proposed method were not compared to any other previously used method to justify its applicability. Some explanations are missing, and some just come up without prior justification. In light of these, I recommend the rejection of this paper. 

Response - About the fact that the results of the proposed method were not compared to any other previously known method, we would like to clarify that the method compared to other modeling tools was not the aim of this work, considerating that the priority was the comparison between output results from the model against observed data, i.e., production data weighted directly from the experimental fields (in this case, the areas where experiments were carried out for the model development) and commercial areas (in this case, large areas of commercial fields where the model was validated). As indicated by the results of this study, the overlapping between observed and predicted data during development and validation was consistent and accomplished the proposed objective of this manuscript. 

Response - Regarding the comment about the lack of adequate and updated references, we would like to point out that this issue was addressed by quoting new references through the manuscript. All of them are indicated in red in the manuscript text and reference list.

1- The introduction section lacks sufficient references. Suitable references are required for sentences in lines 36-39, 57-60, and 74-76. 

Response 1 – We corrected it by including new references as suggested.

Lines 36-39: We included Vandenberghe et al. 2022; Cursi (2022)

Lines 57-60: We included Hoffman and Todd 2013; Simoes et al. (2005)

Lines 74-76: We included Sumesh et al. (2021); Akbarian et al. (2022)

2- In the third paragraph of the introduction section, the use of vegetation indices in different applications is mentioned. I recommend revising this paragraph and providing more recent papers related to sugarcane studies using vegetation indices, especially relevant to your task. In this case, this paragraph also can highlight the place of your research among previously related publications. You can Provide relevant studies with a few explanations of their workflow. 

Response 2 – The paragraph was reviewed and updated references such as Sumesh et al. (2021) and Akbarian et al. (2022) were included. 

3- Please make the fifth paragraph of the introduction section more concise. What is the utility of introducing different methods? It is better only to provide the limitations of previous works and how they should be resolved to justify your proposed method. 

Response 3 - The modification was included as suggested by the reviewer and the Introduction section was shortened.

4- Line 111; The first sentence requires a revision.  

Response 4 – Line 111 was a mistake during text edition in the Latex format. It was suppressed from the text!

5- What are the codes provided in Table 1? 

Response 5 – The codes refers to sugarcane cultivar identification (commercial designations usually used by breeders to identify cultivars). We made it more clear for readers in the legend and table. 

6- Section 2.1 is "Field experiments"; however, it includes explanations about data and vegetation indices. These explanations should be provided separately. So, add a new subsection for data and vegetation indices. 

Response 6 – We reorganized this section according to reviewer’s suggestion by splitting it in two subsections

7- The explanations about data acquisition, data preparation, and processing are not enough. Further explanations are required for both RPAS and Sentinel-2. These explanations are not organized, and it is hard to follow. I recommend introducing the data (both RPAS and Sentinel-2) and providing a thorough explanation of their acquisition and pre-processing, and then providing explanations about the vegetation indices. Also, it is suggested to provide sample RGB images over the sugarcane field using both RPAS (e.g., entire mosaic) and Sentinel-2. 

Response 7 – We have included a subsection with more detail on data acquisition, preparation, and processing (subsection 2.3). RPAS-generated mosaic figures for the experimental fields are shown in figures 2 and 3, while figures 15-20 in the supplemental material illustrate the Sentinel-2 images for all commercial areas assessed during model validation.

Because the size of the experimental plots in fields A and B used for model development were small (54 to 60 m2), an RPAS device was used to generate images with better resolution (2 to 5 cm/pixel). At another hand, during the model validation step, we used areas bigger than 1 ha (10,000 m2), and these areas are suitable to be evaluated by satellite images such as Sentinel-2, which has a spatial resolution of 10 m.

8- Please merge subsections 2.2, 2.3, and 2.4 as "proposed method" subsection. 

Response 8 – We did that

9- Lines 184-186 and 186-188; Please provide suitable references. 

Response 9 - References have been added.

10- Figure 3; Please provide the average value of vegetation indices in each plot. Also, provide a legend so that the colors will be understandable. 

Response 10 – The average value of all vegetation indices is supplemented as additional tables (Table 6 and Table 7) in the Supplementary Information section. However, we apologize because during the manuscript application, some issues occurred and this supplementary information was not available to reviewers. Now we included all supplementary information in the same LaTex file to assure that all information goes in the same file. 

About the color gradient, we add a new illustration (Figure 3) with a color profile to each experimental plot plus a legend with a short explanation about what does mean the color profile.

11- Line 242; "Figures 15 to 21 and Table 7"; There is no supplementary information to this paper. 

Response 11 – As emphasized above, actually there is supplementary information! Unfortunately, when it was uploaded as an additional file during the manuscript submission, it was not included in the PDF version generated for reviewers. Now, we are including the supplementary information in the same LaTex file of the manuscript.

12- Section 3; It is better only to conserve results and discussion in this section. Therefore, the description of the analysis should have been clearly stated in the methodology section. 

Response 12 – We moved all description of the analysis to the M&M section as suggested by the reviewer

13- It is hard to follow the explanation in Section 3, and many questions arise when reading this section. It requires a thorough revision and re-organization. 

Response 13 - we have rewritten some parts of this section to make it more understandable for readers

14- Figure 5; What are the sources of NDVI and VARI? 

Response 14 – In this case, the scatter diagrams refer to the data generated from Experimental Fields A and B. Therefore, the sources of NDVI and VARI are RPAS orthomosaics  

15- Line 280; It is not clear how you reached this model. Explanations of each step and each component should have been provided explicitly.  

Response 15 –  Note that the predictor variables described in Subsection 3.1 are inserted into the systematic component (Equation 4), in which this systematic component is associated with the parameter i of the inverse Gaussian distribution.

16- Lines 303-305; Please provide a suitable reference. 

Response 16 – We included references to this part of the text

17- Lines 356-358; I don't think your data was large (from Figures). Why have you claimed that your model was versatile to handle large data? How many corresponding observations were used? 

Response 17 – This part of the text was suppressed from the manuscript.

18- Figure 8; What is the index? 

Response18  - The term index was replaced with observations where each observation corresponds to the predicted (blue dots) and measured (red dots) values of sugarcane productivity. We also fix that in the legend of this figure.

19- Section 3.4; Several explanations are provided about the climatic factors. But these are not enough, and deeper explanations are required about the effect of the climatic variations on your model and results. How do they affect your model? 

Response 19 – We suppressed this section from the manuscript because the effect of climate variable were not significative for the modeling during the parameters selection. Therefore the section about climate can be removed without impairing the understanding of the results.

Although it seems strange to say that the effect of climate has no influence on productivity, in our case due to the peculiarities in setting up the experiments iin two locations with different production techniques associated with sugarcane varieties with different behavior, the variability generated by these parameters per se was enough to develop and adjust the model.

20- Line 380; What do you mean by "the real know yields"? Did you use any simulated yield before?

Response 20 - It means the model was validated with measured data of the sugarcane production (values actually weighed by farmers during the field harvesting). Therefore, the model was developed and validated by comparing the predicted production with the corresponding observed production that was measured to all experimental plots and all comercial fields used in this study.

21- Line 382; Please provide the location of Six sites. You can show them in the revised Figure 3. What do you mean by "30 fields"? You have only been introduced to fields: Field A and Field B. 

Response 21 – It is well detailed and illustrated in the Supplementary Information that follows at the end of the manuscript, including the centroids (lat/log) for each location in table 7. Additionally, figures 15 to 20 show all locations with their respective five plots. 

22- Section 4.1; Please see comment 12. 

Response 22 – We did the adjustment of this section

23- Line 410; It is not clear how you reached this model. Explanations of each step and each component should have been provided explicitly. 

Response 23 –  We included a flowchart with each step to reach the statistical  model.

24- Table 4; Where are the results of location A? 

Response 24 – Table 4 shows the results, when we are taking into account Location A as a reference (it is in the intercept). So all the other levels (B,..., F) are being compared against A.

25- Line 424; There is no location A in Table 4. 

Response 25 – The answer is tha same of the previous question.

26- The provided explanations and workflow to reach Table 5 are not clear.

Response 26 – Table 5 refers to the other levels as a reference in the intercept (in addition to level A which is already in Table 4), what we did to arrive at the results in Table 5 was to fix level B and compare it with the others (less with level A which is already in Table 4). We also did this for levels C,...,F.

27- Figure 14; What is the index?

Response 27 - The term index was replaced with observations where each observation corresponds to the predicted (blue dots) and measured (red dots) values of sugarcane productivity. We also fix that in the legend of this figure.

Reviewer 2 Report

This article presents a systematic study on two sugarcane fields and develops three methods (NDVI, VARI, inverse gaussian) for evaluating sugarcane with various factors. The analysis statements are quite clear, and the experimental research is sufficient to support the model validation. The reviewer has a few questions and hopes the authors can consider that carefully through this revision.

1) Please double-check if anything needs to be modified in Section 2, line 111.

2) When setting the two experimental fields, four types of sugarcane were planted; what are the differences among these four types? Do they have a consistent growth cycle period so that their NDVI and VARI indices perform similarly in testing?

3) As shown in Fig 4, the productivity between CV_0618 and CV_7870 is different in the two fields, where CV_0618 is the lowest in Field B and CV_7870 is the lowest in Field A. Could you explain the possible reasons?

4) The reviewer suggests clearer scales to represent the x-axis in Fig. 9-b,d rather than only using the first letter of each month.

Author Response

Response to Reviewer 2 Comments

Dear reviewer, first of all, we thank you for your comments and contributions to improving our manuscript. All adjusted and corrected points are indicated as red color in the manuscript. To each reviewed point was added an explanation or justification as requested.

1) Please double-check if anything needs to be modified in Section 2, line 111.

Response 1 – It was a mistake in the LaTex file it is fixed in the new manuscript verssion!

2) When setting the two experimental fields, four types of sugarcane were planted; what are the differences among these four types? Do they have a consistent growth cycle period so that their NDVI and VARI indices perform similarly in testing?

Response 2 – As indicated in Table 1, the cycle of each variety is described. We choose these varieties because they have different growth behavior and therefore we access more variability for this crop either for cane production or VIs measurements.

3) As shown in Fig 4, the productivity between CV_0618 and CV_7870 is different in the two fields, where CV_0618 is the lowest in Field B and CV_7870 is the lowest in Field A. Could you explain the possible reasons?

Response 3 – In Brazil, the cultivation of sugarcane is usually classified in different environments of production based mainly on the soil fertility and texture (sandy, silty, or clayey), rainfall distribution, etc. Therefore, different cultivars show different performances in different environments. Sometimes a specific cultivar grows well in a particular environment but may perform very poorly in other environments. That was the case for the different behavior of CV0618 and CV7870 in fields A and B. That was the reason we decided to included more than one sugarcane variety in our study! Therefore, we was able to evaluate the influence of variability of these genotypes in relation to the model.

4) The reviewer suggests clearer scales to represent the x-axis in Fig. 9-b,d rather than only using the first letter of each month.

Response 4 – We removed the whole section of Climate of the manuscript according to other reviewer suggestion, including Figure 9, since it was not directly related to the model development and results.

Reviewer 3 Report

Development and validation of a model based on vegetation indices for the prediction of sugarcane yield

In this study, we compared vegetation indices calculated from images directly collected in experimental and commercial fields in which the sugarcane yield was precisely recorded. A semiparametric model based on the inverse Gaussian distribution was developed and used to analyze data for biometric and agronomic traits of sugarcane. As a result, we were able to design and validate a model for vegetation indices that showed significant overlap between observed and predicted data points that can therefore be used to forecast sugarcane yield with sufficient accuracy. Thus, this model is a versatile statistical tool with direct applications to sugarcane crops.

The work is relevant to the scope of journal. In my opinion, the article is publishable after addressing the given changes. 

Please directly start abstract with main objectives of the study. 

Write empirical results in the abstract. 

The literature reviewed should be from international studies. Therefore, I highly recommend adding the given statement in the introduction section with given studies [1-5]. In particular, the first sentence of fourth paragraph of introduction should have to update as “Classical linear regression models are widely used in various fields of study to estimate the nexus between exogenous and endogenous variables [1-5]”

[1] Estimating smart energy inputs packages using hybrid optimisation technique to mitigate environmental emissions of commercial fish farms

[2] Extreme weather events risk to crop-production and the adaptation of innovative management strategies to mitigate the risk: A retrospective survey of rural Punjab, Pakistan.

[3] Access to output market by small farmers: The case of Punjab, Pakistan

[4] Sensitivity analysis of greenhouse gas emissions at farm level: case study of grain and cash crops

[5] Analysis of Energy Input–Output of Farms and Assessment of Greenhouse Gas Emissions: A Case Study of Cotton Growers

*Authors must have to update the above statement with adding given studies/reference [1-5] in the revised paper. 

Please write the main contributions of the article in the bullet points at the end of introduction section. 

Please add the research questions at the end of introduction section. 

You must have to rephrase the line 111. 

All results should have to justify with previous studies. 

Please write the main policy implications at the end of conclusion section. 

English should have to correct throughout the article.

Author Response

Response to Reviewer 3 Comments

Dear reviewer, first of all, we thank you for your comments and contributions to improving our manuscript. All adjusted and corrected points are indicated in red text. To each reviewed point was added an explanation or justification.

Point 1 - Please directly start abstract with main objectives of the study.  

Response 1 - The abstract was adjusted according to the suggestion.

Point 2 - Write empirical results in the abstract.  

Response 2 - The abstract was adjusted according to the suggestion.

Point 3 - The literature reviewed should be from international studies. Therefore, I highly recommend adding the given statement in the introduction section with given studies [1-5]. In particular, the first sentence of fourth paragraph of introduction should have to update as “Classical linear regression models are widely used in various fields of study to estimate the nexus between exogenous and endogenous variables [1-5]”

[1] Estimating smart energy inputs packages using hybrid optimisation technique to mitigate environmental emissions of commercial fish farms

[2] Extreme weather events risk to crop-production and the adaptation of innovative management strategies to mitigate the risk: A retrospective survey of rural Punjab, Pakistan. 

[3] Access to output market by small farmers: The case of Punjab, Pakistan - [4] Sensitivity analysis of greenhouse gas emissions at farm level: case study of grain and cash crops 

[5] Analysis of Energy Input–Output of Farms and Assessment of Greenhouse Gas Emissions: A Case Study of Cotton Growers 

*Authors must have to update the above statement with adding given studies/reference [1-5] in the revised paper.  

Response 3 - We kindly appreciate the suggestions made by the reviewer. As requested by other reviewers too, we did a throughout adjustment to the references cited in this manuscript with the inclusion of several up-to-date scientific articles more related and focused in orbital and suborbital imagery applied for crop evaluation and model development.

Concerning the indication to use international studies, we would like to stress that Brazil is the largest sugarcane producer in the world and, therefore, Brazil has a key role as a developer of science and technology for this crop. Thus, references whose authors are affiliated with this country are reliable and up to date. Besides that, the majority of references cited in this manuscript are of international scope!

Point 4 - Please write the main contributions of the article in the bullet points at the end of introduction section.  

Response 4 - The following bullet points were add in the text accordling to the reviwer sugestion:

1- RGB and multispectral images of sugarcane fields obtained by drones or satellites are useful source of data to evaluate the sugarcane growth and biomass accumulation;

2- Adjustment of semiparametric regression models applied to vegetative indices allows the accurate prediction of sugarcane traits such as sugarcane yield;

3- the use of remotely collected aerial imagery can be used as an alternative to add or even replace laborious, expensive and time-consuming practices in sugarcane fields;

4- The model of sugarcane yield prediction demonstrated good ability to overlapping predicted data with observed data

Point 5 - Please add the research questions at the end of introduction section. 

Response 5 - We add it to the end of Introduction Section

Point 6 - You must have to rephrase the line 111.  

Response 6 – It was corrected in the LaTex file and removed from the text.

Point 7 - All results should have to justify with previous studies.  

Response 7  – Although we recognize that it is important to compare our results with results previously obtained by other research groups, in this case, due to the novelty of the adopted approach, there are no references available at this time to justify the results obtained. However, in the introduction section there are mention to works that used semiparametric models for other cultures, addressing other aspects of production.

Point 8 - Please write the main policy implications at the end of conclusion section.  

Response 8 – We included it to the end of Conclusion Section

Point 9 - English should have to correct throughout the article. 

Response 9 – We did that by using a professional English editoration service. The Certificate of English edition from AJE is provided as attached file.

Round 2

Reviewer 1 Report

I appreciate the authors' efforts to consider my comment and improve the quality of the paper. Most comments were addressed, but few necessary comments remained and should be addressed thoroughly. 

1- Regarding comparing the method with previously published methods, the authors replied, "We would like to clarify that the method compared to other modeling tools was not the aim of this work". Indeed it is accepted that the model was validated using in-situ observations, as a necessary step, and obtained acceptable results. However, as a new method, its capability should be compared with previously known methods to justify its suitability. Previously published methods also obtained acceptable results when compared to in-situ observations. Therefore, a comparison is required to illustrate that the proposed method has some advantages (e.g., accuracy, computation time, etc.) over previously published papers. 

2- The limitations of this work, along with comparisons, should be included in a separate discussion section.

3- Regarding the impact of climatic factors, the authors stated, "We suppressed this section from the manuscript because the effect of climate variable were not significative for the modeling during the parameters selection". First, as I checked the earlier version of the paper, nothing was mentioned about the insignificant impact of climatic factors, and the explanations sound to be logical. Therefore, explanations about the impact of climatic factors on the production yield of different types of sugarcane and the provided models should be provided thoroughly. The statistical results are required to support the insignificance/significance of the impact of the climatic factors.

4- Several comments were just answered in the response letter. The aim of such comments is to increase the readability of the paper for all interested scholars, and they should be applied to the paper. The captions of Figures/Tables should be clearly written. As per comment 14 (regarding Figure 5), the authors explicitly determined the source of data, but the caption was not correctly revised. Please, clearly state the source of data (e.g., RPAS NDVI, etc.) in all Figures and Tables. Please, take this into account about all relevant Figures/Tables.

5- Please use consistent fonts and style for legends in all Figures (e.g., Figure 3). 

Author Response

We would like to thank the reviewer for their thoughtful comments and efforts toward improving our manuscript.
Attending the reviewer requestions, below follows the response to each highlighted question:
1- We included a new topic for comparison of the model proposed in this manuscript with a standard model based on multilinear regression. Results from this comparison are showed as a table and they are discussed throughout the text.  
2 - We created a new subsection to address this topic.
3 - We returned the weather data to the manuscript and explained why this data was not used in the model generation.
4 - As requested, we fix it by adding extra information to the captions.
5 - As requested, we addressed this issue and improved the quality of all legends.

Reviewer 3 Report

Appreciate efforts from authors in improving the manuscript. Authors have been answered all my questions, I have no further question. 

Author Response

We would like to thank the reviewer for their thoughtful comments and efforts toward improving our manuscript.

Round 3

Reviewer 1 Report

Thanks for considering the comments to improve the quality of the paper. Three minor issues should be implemented before the publication of the paper.

1- The captions of Figures/Tables are still incomplete, even after two rounds of revision. Please explicitly state the source of data (e.g., RPAS or Sentinel-2) in captions. For instance, what is the source of NDVI and VARI in Figure 14 (and all other relevant Figures/Tables)? As another example, What was the source of prediction in Figure 15?

2- I did not find the new subsection regarding the authors' reply: "We created a new subsection to address this topic". It is related to stating the limitation(s) of the proposed method.

3- What is the relevance of sugarcane yield estimation with apple yield prediction? (Lines 541-546). I don't get the point of comparison.

Author Response

Dear reviewer

Thank you for all of your suggestions and the careful reading of our manuscript.  

We hope the improvement met all demands!

About the required corrections and adjustments, following below is the response to each one of them:

1- The captions of Figures/Tables are still incomplete, even after two rounds of revision. Please explicitly state the source of data (e.g., RPAS or Sentinel-2) in captions. For instance, what is the source of NDVI and VARI in Figure 14 (and all other relevant Figures/Tables)? As another example, What was the source of prediction in Figure 15?

Response - We fix it and it is indicated in red color in the manuscript text.

2- I did not find the new subsection regarding the authors' reply: "We created a new subsection to address this topic". It is related to stating the limitation(s) of the proposed method.

Response - We add the subsection (4.3. Model comparison). It is indicated in red color in the manuscript text.

3- What is the relevance of sugarcane yield estimation with apple yield prediction? (Lines 541-546). I don't get the point of comparison.

Response - We replace the reference with apple yield prediction for a more suitable reference using as a model sugarcane crop to predict yield. It is also indicated in the red color in the manuscript text. 

Yours sincerely
